# What Matters in
# Range View 3D Object Detection

**Benjamin Wilson**
Georgia Institute of Technology
benjaminrwilson@gatech.edu

**Nicholas Autio Mitchell**
University of Freiburg
naumitch@cs.uni-freiburg.de

**Jhony Kaesemodel Pontes**
Latitude AI
jpontes@lat.ai

**James Hays**
Georgia Institute of Technology
hays@gatech.edu

**Abstract:** Lidar-based perception pipelines rely on 3D object detection models to interpret complex scenes. While multiple representations for lidar exist, the range view is enticing since it losslessly encodes the entire lidar sensor output. In this work, we achieve state-of-the-art amongst range view 3D object detection models without using multiple techniques proposed in past range view literature. We explore range view 3D object detection across two modern datasets with substantially different properties: Argoverse 2 and Waymo Open. Our investigation reveals key insights: (1) input feature dimensionality significantly influences the overall performance, (2) surprisingly, employing a classification loss grounded in 3D spatial proximity works as well or better compared to more elaborate IoU-based losses, and (3) addressing non-uniform lidar density via a straightforward range subsampling technique outperforms existing multi-resolution, range-conditioned networks. Our experiments reveal that techniques proposed in recent range view literature are not needed to achieve state-of-the-art performance. Combining the above findings, we establish a new state-of-the-art model for range view 3D object detection — improving AP by 2.2% on the Waymo Open dataset while maintaining a runtime of $10\,\mathrm{Hz}$. We are the first to benchmark a range view model on the Argoverse 2 dataset and outperform strong voxel-based baselines. All models are multi-class and open-source. Code is available at https://github.com/benjaminrwilson/range-view-3d-detection.

**Keywords:** 3D Object Detection, 3D Perception, Autonomous Driving

## 1 Introduction

Lidar-based 3D object detection enhances how machines perceive and navigate their environment — enabling accurate tracking, motion forecasting, and planning. Lidar data can be represented in various forms such as unordered points, 3D voxel grids, bird's-eye view projections, and range view representations. Each representation differs in terms of its sparsity and how it encodes spatial relationships between points. Point-based representations preserve all information but compromise on the efficient computation of spatial relationships between points. Voxel-based and bird's-eye view representations suffer from information loss and sparsity, yet they maintain efficient neighborhood computation. The range view representation preserves the data losslessly and densely in the "native" view of the sensor, but 2D neighborhoods in such an encoding can span enormous 3D distances and objects exhibit scale variance because of the perspective viewpoint.

The field of range-view-based 3D object detection is relatively less explored than alternative representations. Currently, the research community focuses on bird's-eye view or voxel-based methods.

8th Conference on Robot Learning (CoRL 2024), Munich, Germany.

This is partially due to the performance gap between these models and range-view-based models. However, we speculate that the lack of open-source range view models prevents researchers from easily experimenting and innovating within this setting [1, 2, 3, 4]. Our research reveals several unexpected discoveries, including: (1) input feature dimensionality significantly influences overall performance in 3D object detection in the range view by increasing network expressivity to capture object discontinuities and scale variance, (2) a straightforward classification loss based on 3D spatial proximity yields superior generalization across datasets compared to intricate 3D Intersection over Union (IoU)-based losses, (3) simple range subsampling outperforms complex, range-specific network designs, and (4) range view 3D object detection can be competitive across multiple datasets. Surprisingly, we find *without* including certain contributions from prior work, we end up with a straightforward, 3D object detection model that pushes state-of-the-art amongst range view models on both the Argoverse 2 and Waymo Open datasets.

While the goal of this work is *not* to set a new state-of-the-art in 3D object detection, we show that range view methods are still competitive without the need for "bells and whistles" such as model ensembling, time aggregation, or single-category models.

Our contributions are outlined as follows:

1. **Analysis of What Matters.** We provide a detailed analysis on design decisions in range view 3D object detection. Our analysis shows that four key choices impact downstream performance and runtime – input feature dimensionality, 3D input encoding, 3D classification supervision, and range-based subsampling. When these design decisions are optimized, we arrive at a relatively simple range view architecture that is surprisingly competitive with strong baseline methods of any representation.

2. **Simple, Novel Modules.** We propose a straightforward classification loss grounded in 3D spatial proximity yields superior generalization across datasets compared to more complex IoU-based losses [5] which generalizes surprisingly well across the Argoverse 2 and Waymo Open datasets. We introduce a simple, range subsampling technique which outperforms multi-resolution, range-conditioned network heads [5].

3. **High Performance without Complexity.** Surprisingly, without range-specific network designs [5, 2], or IoU prediction [5], we demonstrate that range view based models are competitive with strong voxel-based baselines on the Argoverse 2 dataset and establish a new state-of-the-art amongst range view based 3D object detection models on the Waymo Open dataset — improving L1 mAP by 2.2% while running at 10 Hz.

4. **Open Source, Multi-class, Portable.** Prior range view methods have not provided open-source implementations [1, 2], used single-class detection designs [5], or have been written in non-mainstream deep learning frameworks [5]. We provide multi-class, open-source models written in Pytorch on the Argoverse 2 [6] and Waymo Open [7] datasets with open-source implementations to facilitate range-view-based, 3D object detection research at https://github.com/benjaminrwilson/range-view-3d-detection.

## 2 Related Work

**Point-based Methods.** Point-based methods aim to use the full point cloud without projecting to a lower dimensional space [8, 9, 10]. PointNet [8] and Deep Sets [11] introduced *permutation invariant* networks which enabled direct point cloud processing, which eliminates reliance on the relative ordering of points. PointNet++ [9] further improved on previous point-based methods, but still remained restricted to the structured domain of indoor object detection. While PointRCNN [10] extends methods to the urban autonomous driving domain, point-based methods do not scale well with the number of points and size of scenes, which makes them unsuitable for real-time, low-latency safety-critical applications.

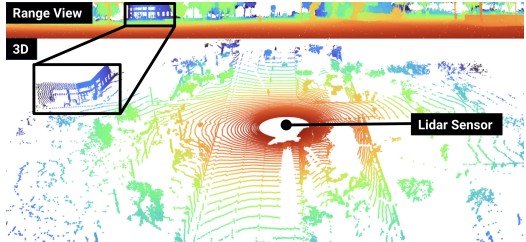
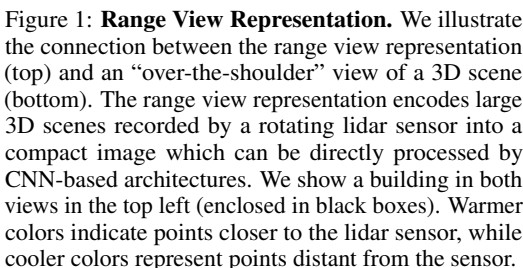

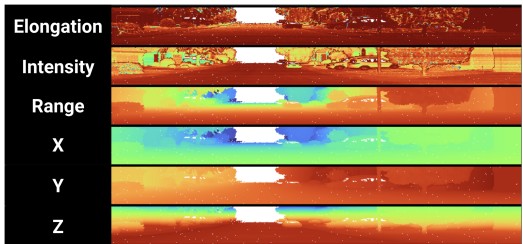

Figure 1: **Range View Representation.** We illustrate the connection between the range view representation (top) and an "over-the-shoulder" view of a 3D scene (bottom). The range view representation encodes large 3D scenes recorded by a rotating lidar sensor into a compact image which can be directly processed by CNN-based architectures. We show a building in both views in the top left (enclosed in black boxes). Warmer colors indicate points closer to the lidar sensor, while cooler colors represent points distant from the sensor.

Figure 2: **Network Inputs: Range View Features.** The input to our network for the Waymo Open dataset consists of auxiliary features (elongation and intensity) and geometric features (range, x, y, z). Each channel is re-mapped to represent warmer colors as the smallest values and cooler colors as the largest values within their respective domains. White pixels indicate invalid returns.

**Grid-based Projections.** Grid-based projection methods first discretize the world into either 2D Bird's Eye View [12, 13, 14, 15] or 3D Voxel [16, 17, 18, 19] grid cells, subsequently placing 3D lidar points into their corresponding 2D or 3D cell. These methods often result in collisions, where point density is high and multiple lidar points are assigned to the same grid cell. While some methods resolve these collisions by simply selecting the nearest point [12], others use a max-pooled feature [14] or a learned feature fusion [19, 13].

**Range-based Projections.** A range view representation is a *spherical* projection that maps a point cloud onto a 2D image plane, with the result often referred to as a *range image*. Representing 3D points as a range image yields a few notable advantages: (a) memory-efficient representation, i.e., the image can be constructed in a way in which few pixels are "empty", (b) implicit encoding of occupancy or "free-space" in the world, (c) compute-efficient — image processing can be performed with dense 2D convolution, (d) scaling to longer ranges is invariant of the discretization resolution. A range image can be viewed as a dense, spherical indexing of the 3D world. Notably, spherical projections are $O(1)$ in space complexity as a function of range at a *fixed* azimuth and inclination resolution. Due to these advantages, range view has been adapted in several works for object detection [1, 20, 21, 5, 3]. Subsequent works have explored the range view for joint object detection and forecasting [22, 23].

**Multi-View Projections.** To leverage the best of both projections, recent work has explored multi-view methods [19, 13, 15, 24]. These methods extract features from both the bird's-eye view and the range view and fuse them to generate object proposals. While Zhou *et al.* [19] proposes to fuse multi-view features in the point space, Laddha *et al.* [13] suggests fusing these features by projecting range view features to the bird's-eye view. Fadadu *et al.* [15] investigates the fusing of RGB camera features with multi-view lidar features. In this work, we explore how competitive range view representations can be without these additional views.

## 3 Range View 3D Object Detection

We begin by describing the range view and its features, then we outline the task of 3D object detection and its specific formulation in the range view. Next, we describe a common architecture for a range-view-based 3D object detection model.

**Inputs: Range View Features.** The range view is a dense grid of features captured by a lidar sensor. In Fig. 1, we illustrate the range view, its 3D representation along with an explicit geometric correspondence of a building between views and the location of the physical lidar sensor which

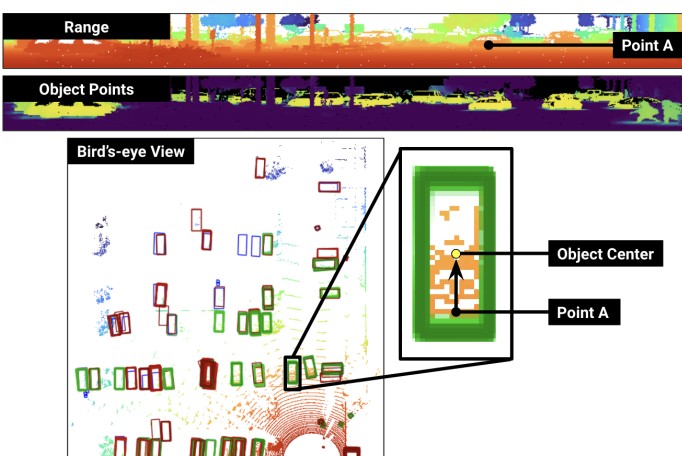

Figure 3: **3D Object Detection in the Range View.** We show a range image, the object confidences from a network, and their corresponding 3D cuboids shown in the bird's-eye view for a scene with multiple parked vehicles. For each visible point in the range image, our range view 3D object detection model learns (1) which category an object belongs to (2) the offset from the visible point to the center of the object, its 3D size, and its orientation. In the above example, we show one particular point (Point A) from two different perspectives — the range view and the bird's-eye view. Blue boxes indicate the ground truth cuboids, green boxes indicate true positives, and red boxes indicate false positives. Importantly, each object can have many thousands of proposals — however, most will be removed through non-maximum suppression.

captures the visualized 3D data. Fig. 2 shows the range view features used for our Waymo Open experiments.

**3D Object Detection.** Given range view features shown in Fig. 2, we seek to map over 50,000 3D points to a much smaller set of objects in 3D and describe their location, size, and orientation. Given the difficulty of this problem, approaches usually "anchor" their predictions on a set of 3D "query" points (3D points which are pixels in range view features). For each 3D point stored as features in a range image, we predict a 3D cuboid which describes a 3D object. Fig. 3 illustrates the range view input (top), the smaller set of object points (middle), and the regressed object cuboids prior to non-maximum suppression (bottom). Importantly, each object may contain thousands of salient points which are transformed into 3D object proposals.

**3D Input Encoding.** Feature extraction on 2D grids is a natural candidate for 2D-based convolutional architectures. However, unlike 2D architectures, a range view architecture must reason in *3D* space. Prior literature has shown that learning this mapping is surprisingly difficult [25], which motivates the use of 3D encodings. 3D encodings incorporate explicit 3D information when processing features in the 2D neighborhood of a range image. For example, the Meta-Kernel from RangeDet [5] weights range view features by a relative Cartesian encoding. We include a cross-dataset analysis of two different methods from prior literature in our experiments. Unexpectedly, we discover that not all methods lead to improvements over our baseline encoding.

**Scaling Input Feature Dimensionality.** The backbone stage performs feature extraction for range view features which have been processed by the 3D input encoding. We adopt a strong baseline architecture, Deep Layer Aggregation (DLA), for all of our experiments following prior work [5]. We find that scaling feature dimensionality *significantly* impacts performance across datasets.

**Dynamic 3D Centerness.** We *propose* a dynamic 3D classification supervision method motivated by VarifocalNet [26]. During training, we compute classification targets by computing the spatial proximity between an object proposal and its assigned ground truth cuboid via a Gaussian likelihood:

$$C_{3D}(d_i, g_i) = \exp\left(\frac{-r_i}{\sigma^2}\right), \text{ where } r_i = ||d_i^{xyz} - g_i^{xyz}||_2^2, \tag{1}$$

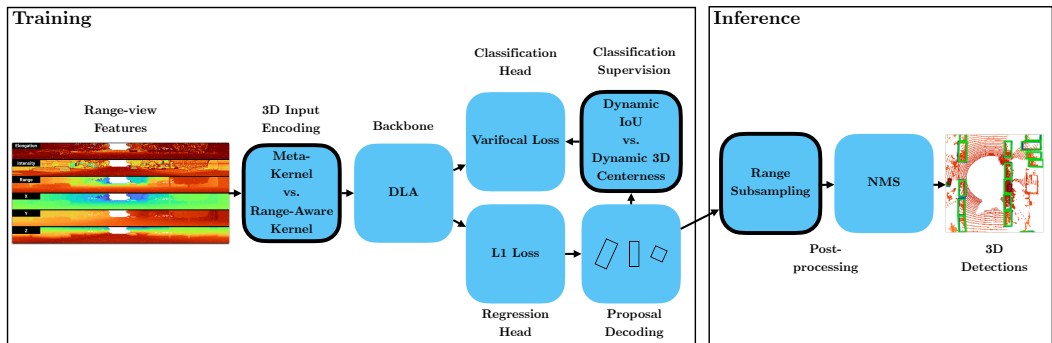

Figure 4: **Model Architecture.** We explore a variety of design decisions in range view based 3D object detection models. Our overall framework is shown above. Range view features are processed by a 3D input encoding which modulates features by their proximity in 3D space. These features are subsequently passed to a backbone CNN for feature extraction and sharing. The classification and regression process these features and produce classification likelihoods and object regression parameters, respectively. The regression parameters are compared with their ground truth target assignments to produce classification targets which incorporate regression-quality. The classification scores and decoded bounding boxes are subsampled by our Range Subsampling method and then finally clustered via non-maximum supression to produce the final set of likelihoods and scores. Blue boxes indicate core components in the network and boxes outlined in black indicate components which we explicitly ablate and explore.

where $d_i^{xyz}$ and $g_i^{xyz}$ are the coordinates of the assigned object proposal and its corresponding ground truth annotation, and $\sigma$ controls the width of the Dynamic 3D Centerness. We adopt $\sigma = 0.75$ for all experiments. Importantly, our Dynamic 3D Centerness method is computed in *3D space*, not pixel space, and it's a function of the object proposals produced during each step of training. We compare our Dynamic 3D Centerness approach to the Dynamic $IoU_{BEV}$ method proposed in prior work [5].

**Range Subsampling.** The non-uniform density of lidar sensors causes nearby objects to have *significantly* more proposals since we make predictions at every observed point — with some objects containing many thousands of points. Processing large numbers of proposals is expensive, but also redundant since nearby objects have many visible points. We propose a straightforward Range Subsampling (RSS) method, which addresses runtime challenges without introducing any additional parameters and simplifies the overall network architecture. For a dense detection output from the network, we partition the object proposals by a set of non-overlapping range intervals. Proposals closer to the origin are subsampled heavily, while proposals at the far range are not subsampled. Despite its simplicity, we will show that it outperforms complex multi-resolution, range-conditioned architectures [5] in our experimental section.

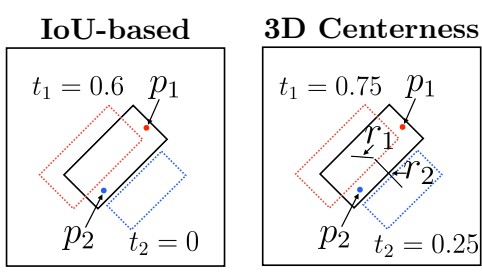

Figure 5: **Dynamic 3D Classification Supervision.** We decode object proposals at each 3D point in a range image during training in order to rank them and compute a soft classification target $t_i$. In the above example, we show two object points, $p_1$ (red) and $p_2$ (blue), their corresponding proposals decoded from the network (color-coded), the soft targets $t_1$ and $t_2$, and the radii computed for Dynamic 3D Centerness, $r_1$ and $r_2$. We illustrate the differences between IoU-based (left) and our proposed Dynamic 3D Centerness (right) rankings. IoU-based metrics are sensitive to translation error and can provide no signal when there is no overlap between the decoded proposal and the ground truth object. Dynamic 3D centerness does not suffer from the same problem.

## 4 Experiments

In this section, we present our experiments on two modern, challenging datasets for range-view-based 3D object detection. Our experi-

ments illustrate which decisions matter when designing a performant range-view-based detection model.

## 4.1 Datasets

**Argoverse 2.** The dataset contains 1,000 sequences of synchronized, multi-modal data. The dataset contains 750 training sequences, 150 validation sequences, and 150 testing sequences. In our experiments, we use the top lidar sensor to construct an $1800 \times 32$ range image. The official Argoverse 2 3D object detection evaluation contains 26 categories evaluated at a 150 m range with the following metrics: average precision (AP), average translation (ATE), scaling (ASE), and orientation (AOE) errors, and a composite detection score (CDS). AP is a VOC-style computation with a true positive defined at 3D Euclidean distance averaged over 0.5 m, 1.0 m, 2.0 m, and 4.0 m. We outline additional information in the appendix and refer readers to Wilson *et al.* [6] for further details.

**Waymo Open.** The Waymo Open dataset [7] contains three evaluation categories: Vehicle, Pedestrian, and Cyclist, evaluated at a maximum range of approximately 80 m. We use the training split (798 logs) and the validation split (202 logs) in our experiments. The dataset contains one medium-range and four near-range lidar sensors. The medium-range lidar sensor is distributed as a single, dense range image. We utilize the $2650 \times 64$ range image from the medium-range lidar for all experiments. We evaluate our Waymo experiments using 3D Average Precision (AP). Following RangeDet [5], we report Level-1 (L1) results. Additional details can be found in the appendix and the original paper [7].

## 4.2 Experiments

In this section, we report our experimental results. Full details on our baseline model can be found in the appendix.

**Input Feature Dimensionality.** We find that input feature dimensionality plays a large role in classification and localization performance. We explore scaling feature dimensionality of the high resolution pathway in both the backbone, the classification, and regression heads. In Table 1, we find that performance on Argoverse 2 consistently improves when doubling the backbone feature dimensionality. Additionally, the error metrics continue to decrease despite increasingly challenging true positives being detected. We report a similar trend in Table 2 on Waymo Open. We suspect that the performance improvements are largely due to learning difficult variances found in the range view (*e.g.* scale variance and large changes in depth). We choose an input feature dimensionality of 256 and 128 for our state-of-the-art comparison on Argoverse 2 and Waymo Open, respectively, to balance performance and runtime.

| $d$ | mAP $\uparrow$ | ATE $\downarrow$ | ASE $\downarrow$ | AOE $\downarrow$ | CDS $\uparrow$ | Latency (ms) $\downarrow$ | |
| --- | --- | --- | --- | --- | --- | --- | --- |
| | | | | | | Backbone | Head |
| 64 | 16.3 | 0.83 | 0.50 | 1.28 | 12.3 | 10.55 | 3.60 |
| 128 | 18.7 | 0.76 | 0.47 | 1.19 | 14.3 | 14.91 | 8.50 |
| 256 | 20.2 | 0.64 | 0.41 | 0.98 | 15.6 | 26.20 | 24.52 |
| 512 | **21.0** | **0.47** | **0.35** | **0.89** | **16.3** | 58.06 | 82.88 |

Table 1: **Input Feature Dimensionality: Argoverse 2.** Evaluation metrics across four input feature dimensionalities $d$ shown on the Argoverse 2 *validation* set. Scaling the high resolution feature dimensionality of the network in both the backbone and head leads to substantial performance improvements — increase from 16.3% to 21.0% mAP while reducing true positive errors.

| $d$ | 3D AP$_{L1}$ $\uparrow$ | | | Latency (ms) $\downarrow$ | |
| --- | --- | --- | --- | --- | --- |
| | Vehicle | Pedestrian | Cyclist | Backbone | Head |
| 64 | 60.34 | 68.57 | 28.47 | 24.74 | 9.19 |
| 128 | **64.00** | **71.77** | **42.37** | 40.31 | 23.98 |

Table 2: **Input Feature Dimensionality: Waymo Open.** Level-1 Mean Average Precision across two input feature dimensionalities $d$ on the Waymo Open *validation* set. Using a larger input feature dimensionality leads to a notable improvement across categories. Further scaling was limited by available GPU memory.

**3D Input Encoding.** Close proximity of pixels in a range image does not guarantee that they are close in Euclidean distance. Previous literature has explored incorporating explicit 3D information to better retain geometric information into the input encodings. We re-implement four of these methods: the Meta-Kernel from RangeDet [5], the Range-aware Kernel (RAK) from RangePerception

[2], PointNet [8], and EdgeConv [27]. We find that the Meta-Kernel outperforms our baseline by 2.2% mAP and 1.9% CDS. Additionally, the average translation and orientation errors are reduced. Unexpectedly, we are unable to reproduce the performance improvement from the Range Aware Kernel. On the Waymo Open dataset, we find that the Meta-Kernel yields a 3.57% and 5.03% improvement over the Vehicle and Pedestrian categories. Consistent with our results on Argoverse 2, the Range Aware Kernel fails to reach our baseline performance. Full results are in the appendix. We will adopt the Meta-Kernel for our state-of-the-art comparison.

**Dynamic 3D Classification Supervision.** We compare four different strategies across the Argoverse 2 and Waymo datasets. In Fig. 5, we illustrate the difference between IoU-based methods and our proposed Dynamic 3D Centerness. On Argoverse 2, our Dynamic 3D centerness outperforms $IoU_{BEV}$ by by 1.9% mAP and 1.3% CDS. We speculate that this performance improvement occurs because Argoverse 2 contains many small objects *e.g.* bollards, construction cones, and construction barrels, which receive low classification scores due to translation error under IoU-based metrics. Dynamic 3D Centerness also incurs less translation, scale, and orientation errors than competing rankings. The optimal ranking strategy remains less evident for the Waymo dataset. The official Waymo evaluation uses vehicle, pedestrian, and cyclist as their object evaluation categories, which are larger on average than many of the smaller categories in Argoverse 2. We find that Dynamic $IoU_{BEV}$ and Dynamic 3D Centerness perform similarly at 60.85% and 60.30% AP; however, for smaller objects such as pedestrian, Dynamic 3D Centerness outperforms $IoU_{BEV}$ by 1.1%. Full tables are in the appendix. Our experiments suggest that IoU prediction is *unnecessary* for strong performance on either dataset. We adopt Dynamic 3D Centerness for our state-of-the-art comparison since it performs well on both datasets.

**Range-based Sampling.** We compare our baseline architecture (single resolution prediction head with no sub-sampling) against the Range-conditioned Pyramid (RCP) [5] and our Range Subsampling (RSS) approach. In Table 3, we surprisingly find that RCP performs nearly *identical* to our baseline model in overall performance; however, it yields a modest improvement in runtime by reducing the total number of object proposals processed via NMS. By sampling object proposals as a post-processing step, our method, RSS, performs on par with RCP and the baseline with no additional parameters or network complexity, and comparable runtime. Similarly, we examine the impact of range-based sampling across the Waymo Open dataset in Table 4. We find that the Range-conditioned pyramid performs *worse* than our baseline despite having 2.8x the number of parameters in the network heads. We speculate that feature-pyramid-network (FPN) approaches are not as effective in the range view since objects cannot be normalized in the manner proposed by the original FPN [28]. We will adopt RSS in our state-of-the-art comparison.

| Method | Head Params. (M) | mAP $\uparrow$ | ATE $\downarrow$ | ASE $\downarrow$ | AOE $\downarrow$ | CDS $\uparrow$ | FPS $\uparrow$ |
|--------|------------------|------|------|------|------|------|------|
| Baseline | **1.2** | 16.2 | 0.82 | 0.50 | 1.28 | 12.3 | 19.51 |
| RCP [5] | 3.4 | 15.9 | 0.83 | 0.51 | 1.28 | 12.0 | 26.0 |
| RSS (ours) | **1.2** | **16.3** | 0.83 | 0.50 | 1.28 | 12.3 | **27.49** |

Table 3: **Subsampling by Range: Argoverse 2.** We compare different subsampling strategies on the Argoverse 2 *validation* set. The range-conditioned pyramid *modifies* the network architecture with each multi-resolution head responsible for a range-interval. In contrast, our RSS approach is a parameter free approach which uses a single-resolution head and only changes the subsampling procedure before NMS.

**Comparison against State-of-the-Art.** Combining a scaled input feature dimensionality with Dynamic 3D Centerness and Range-based sampling yields a model which is competitive with existing voxel-based methods on the Argoverse 2 dataset, and state-of-the-art amongst range view models on the Waymo Open dataset. In Table 5, we report our mAP over the 26 categories in Argoverse 2. We outperform VoxelNext [29], the strongest voxel-based model, by 2.2% mAP with comparable runtime. In Table 6, we show our L1 AP against a variety of different models on Waymo Open. Our method outperforms all existing range view models while also being multi-class.

| Method | Head Params. (M) | 3D AP$_{L1}$ ↑ | | | FPS ↑ |
|---|---|---|---|---|---|
| | | Vehicle | Pedestrian | Cyclist | |
| Baseline | **1.2** | 60.09 | 67.46 | **28.55** | 4.87 |
| RCP [5] | 3.4 | 58.97 | 66.91 | 16.74 | 14.08 |
| RSS (ours) | **1.2** | **60.34** | **68.57** | 28.47 | **19.3** |

Table 4: **Subsampling by Range: Waymo Open.** We compare different subsampling strategies on the Waymo Open *validation* set. RSS outperforms all approaches in both performance and runtime while requiring fewer parameters.

| | Mean | R. Vehicle | Pedestrian | Bollard | C. Barrel | C. Cone | S. Sign | Bicycle | L. Vehicle | B. Truck | W. Device | Sign | Bus | V. Trailer | Truck | Motorcycle | T. Cab | FPS |
|---|---|---|---|---|---|---|---|---|---|---|---|---|---|---|---|---|---|---|
| Distribution (%) | - | 56.92 | 17.95 | 6.8 | 3.62 | 2.63 | 1.99 | 1.42 | 1.25 | 1.09 | 1.06 | 0.91 | 0.83 | 0.69 | 0.54 | 0.47 | 0.44 | - |
| mAP ↑ | | | | | | | | | | | | | | | | | | |
| CenterPoint [30] | 22.0 | 67.6 | 46.5 | 40.1 | 32.2 | 29.5 | - | 24.5 | 3.9 | 37.4 | - | 6.3 | 38.9 | 22.4 | 22.6 | 33.4 | - | - |
| FSD [31] | 28.2 | 68.1 | 59.0 | 41.8 | 42.6 | 41.2 | - | 38.6 | 5.9 | 38.5 | - | 11.9 | 40.9 | **26.9** | 14.8 | 49.0 | - | - |
| VoxelNext [29] | 30.7 | 72.7 | 63.2 | **53.9** | 64.9 | 44.9 | - | 40.6 | **6.8** | **40.1** | - | 14.9 | 38.8 | 20.9 | 19.9 | 42.4 | - | 18 |
| Ours (64) | 28.8 | 74.4 | 65.9 | 45.6 | 70.8 | 48.4 | 30.0 | 31.4 | 4.3 | 35.6 | 16.9 | 17.5 | 41.0 | 15.9 | 21.1 | 39.3 | 16.7 | 31 |
| Ours (128) | 32.9 | 76.2 | 68.4 | 49.0 | 72.1 | 50.3 | 36.5 | 39.7 | 6.6 | 38.0 | 22.6 | 17.9 | 47.1 | 21.0 | **26.4** | 47.7 | 20.0 | 22 |
| Ours (256) | **34.4** | **76.5** | **69.1** | 50.0 | **72.9** | **51.3** | **39.7** | **41.4** | 6.7 | 36.2 | **23.1** | **20.0** | **48.4** | 24.7 | 24.2 | **51.3** | **21.9** | 10 |

Table 5: **State-of-the-Art Comparison: Argoverse 2.** We compare our range view model against different state-of-the-art, peer-reviewed methods on the Argoverse 2 *validation* dataset. We significantly outperform other methods on small retroreflective objects such as construction barrels and cones. The full table will be available in the appendix.

| | Method | Open-source | Multi-class | 3D AP$_{L1}$ ↑ | | |
|---|---|---|---|---|---|---|
| | | | | Vehicle | Pedestrian | Cyclist |
| **Voxel-based** | | | | | | |
| | SWFormer [32] | | ✓ | **77.8** | **80.9** | - |
| **Multi-view-based** | | | | | | |
| | RSN [20] | | | 75.1 | 77.8 | - |
| **Range-view-based** | | | | | | |
| | To the Point [4] | | | 65.2 | 73.9 | - |
| | RangeDet [5] | ✓ | | 72.85 | 75.94 | 65.67 |
| | RangePerception [2] | | ✓ | 73.62 | 80.24 | 70.33 |
| | Ours | ✓ | ✓ | 75.12 | 80.82 | **75.01** |

Table 6: **Comparison against State-of-the-Art: Waymo Open.** We compare our range view model against different state-of-the-art methods on the Waymo *validation* set. Our model outperforms all other range-view-based methods. To the best of our knowledge, we are the only open-source, multi-class range view method. Not all methods report Cyclist performance and we're unable to compare FPS fairly since we do not have access to their code.

# 5   Conclusion

In this paper, we examine a diverse set of considerations when designing a range view 3D object detection model. Surprisingly, we find that not all contributions from past literature yield meaningful performance improvements. We propose a straightforward dynamic 3D centerness technique which performs well across datasets, and a simple sub-sampling technique to improve range view model runtime. These techniques allow us to establish the first range view method on Argoverse 2, which is competitive with voxel-based methods, and a new state-of-the-art amongst range view models on Waymo Open. Our results demonstrate that simple methods are at least as effective as recently-proposed techniques, and that range view models are a promising avenue for future research.

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

# 6 Appendix

Our supplementary materials cover the following: background on 3D object detection in the range view, additional quantitative results, qualitative results, dataset details, and implementation details for our models.

# 7 Range View Representation

The range view representation, also known as a range image, is a 2D grid containing the spherical coordinates of an observed point with respect to the lidar laser's original reference frame. We define a range image as:

$$r \triangleq \{(\varphi_{ij}, \theta_{ij}, r_{ij}) : 1 \leq i \leq H; 1 \leq j \leq W\}, \tag{2}$$

where $(\varphi_{ij}, \theta_{ij}, r_{ij})$ are the inclination, azimuth, and range, and $H$, $W$ are the height and width of the image. Importantly, the cells of a range image are not limited to containing only spherical coordinates. They may also contain auxiliary sensor information such as a lidar's intensity.

## 7.1 3D Object Detection

Given a range image $r$, we construct a set of 3D object proposals which are ranked by a confidence score. Each proposal consists of a proposed location, size, orientation, and category. Let $\mathcal{D}$ represent the predictions from a network.

$$\mathcal{D} \triangleq \left\{d_i \in \mathbb{R}^8\right\}_{i=1}^K, \text{ where } K \subset \mathbb{N}, \tag{3}$$

$$d_i \triangleq \left\{x_i^{\text{ego}}, y_i^{\text{ego}}, z_i^{\text{ego}}, l_i, w_i, h_i, \theta_i, c_i\right\} \tag{4}$$

where $x_i^{\text{ego}}, y_i^{\text{ego}}, z_i^{\text{ego}}$ are the coordinates of the object in the ego-vehicle reference frame, $l_i, w_i, h_i$ are the length, width, and height of the object, $\theta_i$ is the counter-clockwise rotation about the vertical axis, and $c_i$ is the object likelihood. Similarly, we define the ground truth cuboids as:

$$\mathcal{G} \triangleq \left\{g_i \in \mathbb{R}^8\right\}_{i=1}^M, \text{ where } M \subset \mathbb{N}, \tag{5}$$

$$g_i \triangleq \left\{x_i^{\text{ego}}, y_i^{\text{ego}}, z_i^{\text{ego}}, l_i, w_i, h_i, \theta_i, q_i\right\}, \tag{6}$$

where $q_i$ is a continuous value computed dynamically during training. For example, $q_i$ may be set to Dynamic 3D Centerness or $\text{IoU}_{\text{BEV}}$. The detected objects, $\mathcal{D}$ are decoded as the same parameterization as $\mathcal{G}$.

$$\mathcal{D} \triangleq \left\{d_k \in \mathbb{R}^8 : c_1 \geq \cdots \geq c_k\right\}_{k=1}^K, \text{ where } K \subset \mathbb{N}, \tag{7}$$

$$d_k \triangleq \left\{x_k^{\text{ego}}, y_k^{\text{ego}}, z_k^{\text{ego}}, l_k, w_k, h_k, \theta_k\right\}. \tag{8}$$

We seek to predict a continuous representation of the ground truth targets as:

$$\mathcal{D} \triangleq \left\{d_k \in \mathbb{R}^8 : c_1 \geq \cdots \geq c_k\right\}_{k=1}^K, \text{ where } K \subset \mathbb{N}, \tag{9}$$

$$g_k \triangleq \left\{x_k^{\text{ego}}, y_k^{\text{ego}}, z_k^{\text{ego}}, l_k, w_k, h_k, \theta_k, c_k\right\}, \tag{10}$$

where $x_k^{\text{ego}}, y_k^{\text{ego}}, z_k^{\text{ego}}$ are the coordinates of the object in the ego-vehicle reference frame, $l_k, w_k, h_k$ are the length, width, and height of the object, $\theta_k$ is the counter-clockwise rotation about the vertical axis, and $c_k$ is the object category likelihood.

**3D Anchor Points in the Range View.** To predict objects, we bias our predictions by the location of *observed* 3D points which are features of the projected pixels in a range image. For all the 3D points contained in a range image, we produce a detection $d_k$.

**Regression Targets.** Following previous literature, we do not directly predict the object proposal representation in Section 7.1. Instead, we define the regression targets as the following:

$$\mathcal{T}(\mathcal{P}, \mathcal{G}) = \{t_i(p_i, g_i) \in \mathbb{R}^8\}_{i=1}^K, \text{ where } K \in \mathbb{N}, \tag{11}$$

$$t_i(p_i, g_i) = \{\Delta x_i, \Delta y_i, \Delta z_i, \log l_i, \log w_i, \log h_i, \sin \theta_i, \cos \theta_i\}, \tag{12}$$

where $\mathcal{P}$ and $\mathcal{G}$ are the sets of points in the range image and the ground truth cuboids in the 3D scene, $\Delta x_i, \Delta y_i, \Delta z_i$ are the offsets from the point to the associated ground truth cuboid in the point-azimuth reference frame, $\log l_i, \log w_i, \log h_i$ are the logarithmic length, width, and height of the object, respectively, and $\sin \theta_i, \cos \theta_i$ are continuous representations of the object's heading $\theta_i$.

**Classification Loss.** Once all of the candidate foreground points have been ranked and assigned, each point needs to incur loss proportional to its regression quality. We use Varifocal loss [26] with a sigmoid-logit activation for our classification loss:

$$\text{VFL}(c_i, q_i) = \begin{cases} q_i(-q_i \log(c_i) + (1 - q_i) \log(1 - c_i)) & \text{if } q_i > 0 \\ -\alpha c_i^\gamma \log(1 - c_i) & \text{otherwise,} \end{cases} \tag{13}$$

where $c_i$ is classification likelihood and $q_i$ is 3D classification targets (*e.g.*, Dynamic IoU$_{\text{BEV}}$ or Dynamic 3D Centerness). Our final classification loss for an entire 3D scene is:

$$\mathcal{L}_c = \frac{1}{M} \sum_{j=1}^N \sum_{i=1}^{|\mathcal{P}_G^j|} \text{VFL}(c_i^j, q_i^j), \tag{14}$$

where $M$ is the total number of foreground points, $N$ is the total number of objects in a scene, $\mathcal{P}_G^j$ is the set of 3D points which fall inside the $j^{\text{th}}$ ground truth cuboid, $c_i^j$ is the likelihood from the network classification head, and $q_i^j$ is the 3D classification target.

**Regression Loss.** We use an $\ell_1$ regression loss to predict the regression residuals. The regression loss for an entire 3D scene is:

$$\mathcal{L}_r = \frac{1}{N} \sum_{j=1}^N \frac{1}{|\mathcal{P}_G^j|} \sum_{i=1}^{|\mathcal{P}_G^j|} \text{L1Loss}(r_i^j, t_i^j), \tag{15}$$

where $N$ is the total number of objects in a scene, $\mathcal{P}_G^j$ is the set of 3D points which fall inside the $j^{\text{th}}$ ground truth cuboid, $r_i^j$ is the predicted cuboid parameters from the network, and $t_i^j$ are the target residuals to be predicted.

**Total Loss.** Our final loss is written as:

$$\mathcal{L} = \mathcal{L}_c + \mathcal{L}_r \tag{16}$$

## 7.2 Argoverse 2

Additional details on the evaluation metrics used in the Argoverse 2.

- **Average Precision (AP)**: VOC-style computation with a true positive defined at 3D Euclidean distance averaged over $0.5\,\text{m}$, $1.0\,\text{m}$, $2.0\,\text{m}$, and $4.0\,\text{m}$.
- **Average Translation Error (ATE)**: 3D Euclidean distance for true positives at $2\,\text{m}$.
- **Average Scale Error (ASE)**: Pose-aligned 3D IoU for true positives at $2\,\text{m}$.
- **Average Orientation Error (AOE)**: Smallest yaw angle between the ground truth and prediction for true positives at $2\,\text{m}$.
- **Composite Detection Score (CDS)**: Weighted average between AP and the normalized true positive scores:

$$\text{CDS} = \text{AP} \cdot \sum_{x \in \mathcal{X}} 1 - x, \text{ where } x \in \{\text{ATE}_{\text{unit}}, \text{ASE}_{\text{unit}}, \text{AOE}_{\text{unit}}\}. \tag{17}$$

We refer readers to Wilson *et al.* [6] for further details.

## 7.3 Waymo Open

Additional details on the evaluation metrics used in the Waymo Open are listed below.

1. **3D Mean Average Precision (mAP)**: VOC-style computation with a true positive defined by 3D IoU. The gravity-aligned axis is fixed.

   (a) **Level 1 (L1)**: All ground truth cuboids with at least five lidar points within them.

   (b) **Level 2 (L2)**: All ground cuboids with at least 1 point and additionally incorporates heading into its true positive criteria.

Following RangeDet [5], we report L1 results.

# 8 Range View 3D Object Detection

**Baseline Model.**   Our baseline models are all multi-class and utilize the Deep Layer Aggregation (DLA) [33] architecture with an input feature dimensionality of 64. In our Argoverse 2 experiments, we incorporate five input features: x, y, z, range, and intensity, while for our Waymo experiments, we include six input features: x, y, z, range, intensity, and elongation. These inputs are then transformed to the backbone feature dimensionality of 64 using a single basic block. For post-processing, we use weighted non-maximum suppression (WNMS). All models are trained and evaluated using mixed-precision with BrainFloat16 [34]. Both models use a OneCycle scheduler with AdamW using a learning rate of 0.03 across four A40 GPUs using Synchronized BatchNorm. Both the Argoverse 2 and Waymo models use standard data augmentation techniques such as: random axis flipping, scaling from 0.95 to 1.05, and rotation between $-\frac{\pi}{4}$ and $\frac{\pi}{4}$. All models in the ablations are trained for 5 epochs on a uniformly sub-sampled fifth of the training set.

**State-of-the-art Comparison Model.**   We leverage the best performing and most general methods from our experiments for our state-of-the-art comparison for both the Argoverse 2 and Waymo Open dataset models. The Argoverse 2 and Waymo Open models use an input feature dimensionality of 256 and 128, respectively. Both models use the Meta-Kernel and a 3D input encoding, Dynamic 3D Centerness for their classification supervision, and we use our proposed Range-Subsampling with range partitions of [0 - 30 m), [30 m, 50 m), [50 m, ∞) with subsampling rates of 8, 2, 1, respectively. For both datasets, models are trained for 20 epochs.

| Method | mAP ↑ | ATE ↓ | ASE ↓ | AOE ↓ | CDS ↑ |
|---|---|---|---|---|---|
| Dynamic IoU$_{BEV}$ [5] | 14.4 | 0.85 | 0.51 | 1.29 | 11.0 |
| Dynamic IoU$_{3D}$ | 13.8 | 0.87 | 0.50 | **1.26** | 10.5 |
| Dynamic BEV Centerness (ours) | 16.2 | **0.76** | **0.48** | 1.30 | 12.2 |
| Dynamic 3D Centerness (ours) | **16.3** | 0.83 | 0.50 | 1.28 | **12.3** |

Table 7: **Classification Supervision: Argoverse 2.** Evaluation metrics and errors using two different classification supervision methods on the Argoverse 2 *validation* set. We observe that our proposed centerness-based methods outperform IoU-based methods. Dynamic 3D Centerness achieves the best performing mAP and CDS with the BEV variant as a close second. Surprisingly, Dynamic 3D centerness outperforms IoU$_{BEV}$ in average translation, scale, orientation errors.

## 8.1 Oracle Experiments

While our current work only focuses on single-lidar-only detection, we do believe that sensor fusion is an important direction to explore. We ran some preliminary oracle experiments which use "ground truth" information during inference to correct for errors made via the network's classification or regression. We find that overall performance improves by roughly 5% when correcting for classification errors and 8% when correcting for regression errors.

| Method | 3D AP$_{L1}$ ↑ | | |
|---|---|---|---|
| | Vehicle | Pedestrian | Cyclist |
| Dynamic IoU$_{BEV}$ [5] | 60.54 | 67.45 | 23.53 |
| Dynamic IoU$_{3D}$ | **60.85** | 67.48 | 20.79 |
| Dynamic BEV Centerness (ours) | 59.92 | 68.56 | 28.45 |
| Dynamic 3D Centerness (ours) | 60.34 | **68.58** | **28.53** |

Table 8: **Classification Supervision: Waymo Open.** Evaluation metrics and errors using two different classification supervision methods on the Waymo Open *validation* set. Our results suggest that centerness-based approaches are a competitive alternative to IoU$_{BEV}$, while being simpler.

| Method | mAP ↑ | ATE ↓ | ASE ↓ | AOE ↓ | CDS ↑ |
|---|---|---|---|---|---|
| Basic Block | 16.3 | 0.83 | 0.50 | 1.28 | 12.3 |
| Range Aware Kernel* [2] | 15.7 | 0.83 | 0.51 | 1.29 | 11.8 |
| PointNet [8] | 16.4 | 0.82 | 0.50 | 1.27 | 12.4 |
| EdgeConv [27] | 16.7 | **0.78** | **0.49** | **1.17** | 12.7 |
| Meta Kernel [5] | **18.5** | 0.80 | 0.50 | 1.21 | **14.2** |

Table 9: **3D Input Encoding: Argoverse 2.** Mean Average Precision using different 3D input feature encodings on the Argoverse 2 *validation* set. *: Code unavailable. Re-implemented by ourselves.

| Method | 3D AP$_{L1}$ ↑ | | |
|---|---|---|---|
| | Vehicle | Pedestrian | Cyclist |
| Basic Block | 60.34 | 68.58 | 28.53 |
| Range Aware Kernel* [2] | 59.84 | 67.69 | 24.62 |
| PointNet [8] | 61.17 | 69.92 | 36.30 |
| EdgeConv [27] | 61.23 | 70.29 | 42.29 |
| Meta Kernel [5] | **63.92** | **73.61** | **52.84** |

Table 10: **3D Input Encoding: Waymo Open.** L1 Average Precision (AP) across three different 3D input feature encodings on the Waymo *validation* set. The Meta Kernel outperforms all methods improving AP considerably across all categories. Surprisingly, the Range Aware Kernel performs worse than our baseline method. *: Code unavailable. Re-implemented by ourselves based on details in the manuscript [2].

| | Mean | R. Vehicle | Pedestrian | Bollard | C. Barrel | C. Cone | S. Sign | Bicycle | L. Vehicle | B. Truck | W. Device | Sign | Bus | V. Trailer | Truck | Motorcycle | T. Cab | Bicyclist | S. Bus | W. Rider | Motorcyclist | Dog | A. Bus | M.P.C. Sign | Stroller | Wheelchair | M.B. Trailer | FPS |
|---|---|---|---|---|---|---|---|---|---|---|---|---|---|---|---|---|---|---|---|---|---|---|---|---|---|---|---|---|
| Distribution (%) | - | 56.92 | 17.95 | 6.8 | 3.62 | 2.63 | 1.99 | 1.42 | 1.25 | 1.09 | 1.06 | 0.91 | 0.83 | 0.69 | 0.54 | 0.47 | 0.44 | 0.38 | 0.2 | 0.18 | 0.16 | 0.15 | 0.1 | 0.08 | 0.06 | 0.05 | 0.0 | - |
| mAP ↑ | | | | | | | | | | | | | | | | | | | | | | | | | | | | |
| CenterPoint [30] | 22.0 | 67.6 | 46.5 | 40.1 | 32.2 | 29.5 | - | 24.5 | 3.9 | 37.4 | - | 6.3 | 38.9 | 22.4 | 22.6 | 33.4 | - | 20.1 | 25.8 | - | 28.6 | - | 8.7 | 27.4 | 0.5 | - | - | - |
| FSD [31] | 28.2 | 68.1 | 59.0 | 41.8 | 42.6 | 41.2 | - | 38.6 | 5.9 | 38.5 | - | 11.9 | 40.9 | **26.9** | 14.8 | 49.0 | - | 33.4 | 30.5 | - | 39.7 | - | **20.4** | 26.4 | 13.8 | - | - | - |
| VoxelNext [29] | 30.7 | 72.7 | 63.2 | **53.9** | 64.9 | 44.9 | - | 40.6 | 6.8 | **40.1** | - | 14.9 | 38.8 | 20.9 | 19.9 | 42.4 | - | 32.4 | 25.2 | - | 44.7 | - | 20.1 | 39.4 | 15.7 | - | - | 18 |
| Ours (64) | 28.8 | 74.4 | 65.9 | 45.6 | 70.8 | 48.4 | 30.0 | 31.4 | 4.3 | 35.6 | 16.9 | 17.5 | 41.0 | 15.9 | 21.1 | 39.3 | 16.7 | 26.8 | 30.0 | 6.1 | 38.3 | 3.5 | 10.6 | 45.9 | 11.3 | 2.2 | 0.0 | 31 |
| Ours (128) | 32.9 | 76.2 | 68.4 | 49.0 | 72.1 | 50.3 | 36.5 | 39.7 | 6.6 | **38.0** | 22.6 | 17.9 | 47.1 | 21.0 | 26.4 | 47.7 | 20.0 | 32.2 | 36.3 | **7.7** | 43.0 | 4.2 | 18.8 | 48.8 | 12.8 | 10.7 | 0.1 | 22 |
| Ours (256) | **34.4** | 76.5 | 69.1 | 50.0 | 72.9 | 51.3 | 39.7 | 41.4 | 6.7 | 36.2 | 23.1 | 20.0 | 48.8 | 24.7 | 24.2 | 51.3 | 21.9 | 35.9 | 42.2 | 6.8 | 45.7 | 9.4 | 20.3 | 43.2 | 18.7 | 14.3 | 0.2 | 10 |

Table 11: **State-of-the-Art Comparison: Argoverse 2 (All categories).** We compare our range view model against different state-of-the-art, peer-reviewed methods on the Argoverse 2 *validation* dataset. This table includes all categories — some which were omitted due to space in the main manuscript.

## 8.2 Qualitative Results

We include qualitative results for both Argoverse 2 and Waymo Open shown in Figs. 6 and 7.

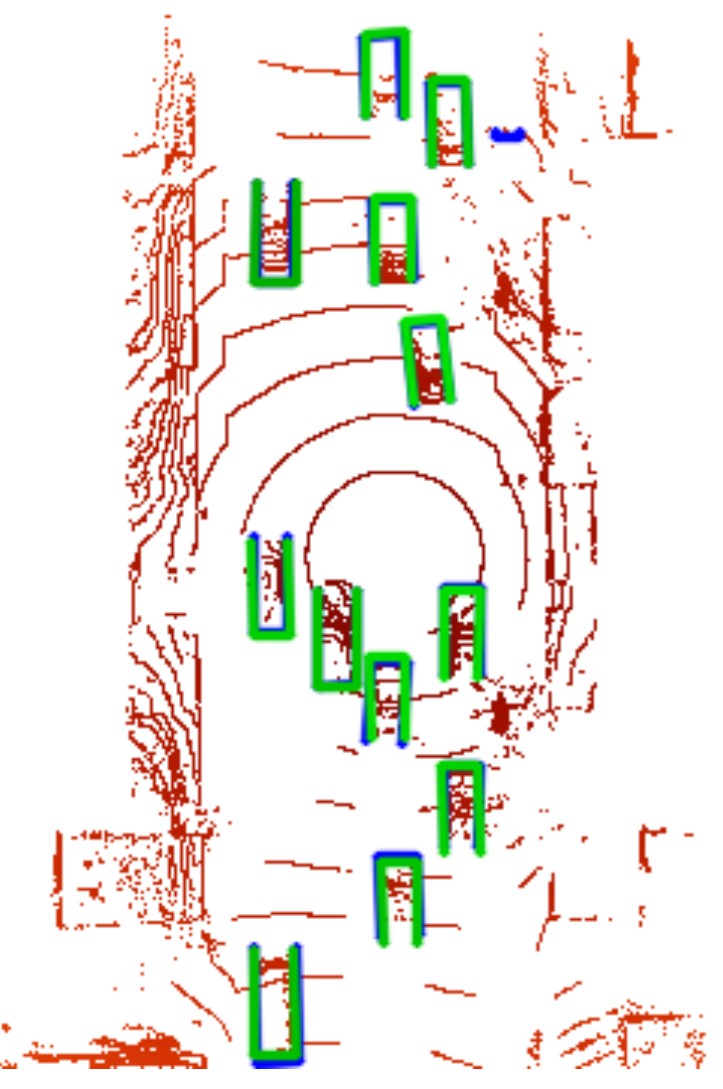

Figure 6: **Qualitative Results: Argoverse 2**. True positives (green) and ground truth cuboids (blue) are shown below for our best performing model. True positives are shown using a $2\,\mathrm{m}$ Euclidean distance from the ground truth cuboid center.

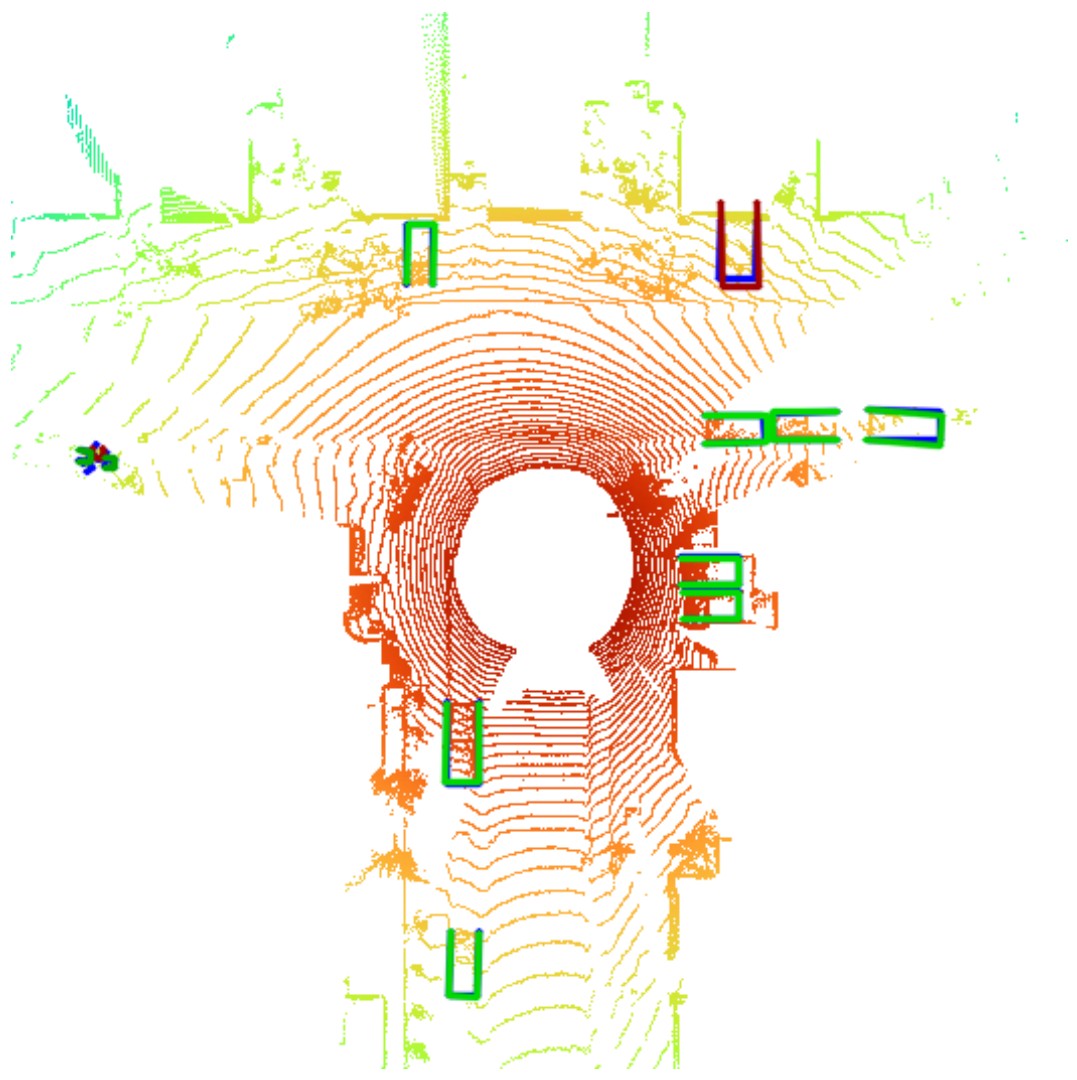

Figure 7: **Qualitative Results: Waymo Open**. True positives (green), false positives (red) and ground truth cuboids (blue) are shown below for our best performing model. True positives are shown using a 0.7 IoU threshold.

