# OpenReview forum: "What Matters in Range View 3D Object Detection"
_robot-learning.org/CoRL/2024/Conference — CoRL 2024_

### Official Review · Reviewer_sbqX · 2024-07-07
**Beat point-based and voxel-based SOTA methods using range-view for the 3D object detection task in Argoverse 2.**

**Originality:** 3
**Technical Quality:** 3
**Clarity Of Presentation:** 4
**Potential Impact:** 3
**Recommendation:** 3
**Confidence:** 4

**Review:**

Quality:
  Strengths:
    1. The code is providied.
    2. Model model deteails, ablation studies, and Qualitative Results visualizations are provided in the supplementary materials.
    3. Experiments are conducted on both Argoverse 2 and Waymo Open Dataset, and the performance is good, especially beats many point-based and voxel-based SOTA on Argoverse 2.
  Weaknesses:
    It would benefit from additional experiments on a more diverse set of datasets to confirm the generalizability of the findings.

Clarity:
  Strengths:
    1. The paper is organized logically, with a clear introduction, and it's well-written and easy to understand.
    2. The figures and tables are clear and easy to understand. (e.g., diagrams illustrating range view representation)
  Weaknesses:
    It would be better to have more types of objects in the visualizations, like pedestrians, cyclists.

Originality:
  Strengths:
    Some novel modules like Dynamic 3D Centerness and Range Subsampling.
  Weaknesses:
    The novel modules are incremental on some exisiting fraemworks, like Meta-Kernel 3D Input Encoding, and Deep Layer Aggregation backbone from RangeDet.

Significance:
  Strengths:
    1. Significant performance improvements for range-view 3D object detection, especially on Argoverse 2, beating many point-based and voxel-based SOTA methods. SOTA for range-view methods.
    2. Provide open-source code for further research and application, enhancing the work's impact on the community.
  Weaknesses:
    1. Some performance improvements are only significant on the Argoverse 2 dataset, like Dynamic 3D Centerness, which lack of generalizability.
    2. I would recommend to provide the model size / runtime of your range-view model, the point-based and voxel-based methods in the performance comparison table, in order to further demonstrate the effeciency of your approach and improve the significance of the paper.

**Quality Of The Limitations Section:**

1

**Questions For Rebuttal:**

Is there some techniqes used in this paper can be used in point-based or voxel-based methods to improve the performance? Essentially in Argoverse 2, that would be a better apple-to-apple comparison or the prove of the significance of the techniqes in this paper.

Is it possible to provide the model size / runtime of the range-view model, the point-based and voxel-based methods in the performance comparison table? To make sure the performance gain is not from the larger model size and make sure they are apple-to-apple comparisons.

Is there any other dataset that can be used to prove the effectiveness of the range-view method?

**Robotics Focus:**

3

**Summary Of Paper:**

In this paper, the authors explore the input feature dimensionality, the classification loss function (3D spatial proximity vs IoU-based), and the way to address non-uniform lidar density (proposed range subsampling vs multi-resolution, range-conditioned networks) to establish a range-view 3D object detection model that outperforms many point-based and voxel-based methods in the Argoverse 2 dataset and Waymo Open Dataset. The authors adopt Deep Layer Aggregation (from RangeDet) as backbone, Varifocal Loss as classification loss, L1 Loss as regression loss, and have ablation studies on Input Feature Dimensionality, 3D Input Encoding methods (Meta-Kernel vs Range-Aware Kernel), Dynamic 3D Classification Supervision (in supplementary materials), and Range-based Sampling.

**Summary Of Recommendation:**

Good performance for range-view 3D object detection especially in Argoverse 2 dataset. The paper is organized logically, with a clear introduction, and it's well-written and easy to understand. Further provement of significance by providing more experiment results is needed. Weak Accept.

---

### Official Review · Reviewer_2jY8 · 2024-07-20
**Initial Review**

**Originality:** 3
**Technical Quality:** 4
**Clarity Of Presentation:** 5
**Potential Impact:** 3
**Recommendation:** 3
**Confidence:** 3

**Review:**

### Strengths
- The paper is well-written and clear.
- The proposed method is simple, yet effective. The overall approach is simpler than prior works, yet outperforms baselines.
- The experimental analysis is rigorous and convincing. The design decisions are motivated via ablation experiments.
- The code is released and has useful improvements (written in PyTorch, multi-class) over existing public codebases for range view detection.


### Weaknesses
- While I believe the simplicity of the approach should be considered a positive, the overall novelty of the technical contributions of this work appear to be minimal. The model architecture is based on prior works and the key technical contributions are (1) a simplified classification loss and (2) a simple sub-sampling technique.
- The details of the sub-sampling technique, one of the core technical contributions of the paper, are not clearly described.
- The limitations of the method and of using range view in general are not well addressed in the paper. E.g., it's not clear how well the method can be extended to support fusing with camera data or supporting multiple lidars on a single vehicle which has become increasing common (e.g., multiple lidars are available in the dataset but only a single one is used)

**Quality Of The Limitations Section:**

2

**Questions For Rebuttal:**

- Can you please share the details the of the sub-sampling technique?

**Robotics Focus:**

3

**Summary Of Paper:**

This paper studies the problem of 3d object detection from a lidar range view representation as input. The paper reviews existing architectures and through experiments demonstrates the modules and designs that most contribute to high performance. The paper then proposed simple alternatives (range subsampling, simplified classification loss)  to existing modules and demonstrates improved performance. Ultimately, the authors propose a simplified architecture is able to achieve state of the art performance.

**Summary Of Recommendation:**

Overall, I believe the technical contributions of the paper would be useful to the community, given that the paper is well-written, technically sound, and presents interesting insights on the problem of range view detection.

---

### Official Review · Reviewer_SCbK · 2024-07-20
**Range View 3D Object Detection Review**

**Originality:** 2
**Technical Quality:** 4
**Clarity Of Presentation:** 4
**Potential Impact:** 2
**Recommendation:** 3
**Confidence:** 3

**Review:**

The paper's premise is straightforward and clearly layed out. As promised in the introduction, the authors perform a series of ablation tests comparing different approaches at each step of their overall method.

Specifically they compare:

- Two existing methods for 3D Input Encoding
- A novel loss function against an existing one for better training
- A novel (simple) subsampling step vs an existing one vs no subsampling at all.

The paper's straightforward premise, clear introduction, and experiments being repeated across two datasets are all strengths. However there are clear weaknesses in the form of:

- Each experiment only comparing two existing methods (three if counting the absence of a method)
- Novel methods proposed are all very straightforward and simple.

The results showing that a simple method such as the one proposed can achieve competitive results is very interesting, and the strongest part of the paper. However it is hard to view that alone sufficient for acceptance. For the other contributions of comparing different steps of the process, more than two methodologies would have to be compared for the paper to be viewed as either a reliable "analysis of what matters" or "measure of the current field".

**Quality Of The Limitations Section:**

1

**Questions For Rebuttal:**

- For the "Analysis of What Matters" contribution why are so few options considered at each step? E.g. in a field filled with countless proposed supervision/loss methods it seems more could easily be considered.
- This lack of options is especially concerning for "3D Input Encoding". Considering only two methods were compared, the fact one of them did not reproduce the claimed results, inevitably leads to the question of whether their paper exaggerated performance, or this paper's authors failed to reproduce it properly. If more methods were compared (and achieved results roughly consistent with their own paper's claims) then a single method performing worse than it's claims would not bring the comparison's reliability into question.

**Robotics Focus:**

2

**Summary Of Paper:**

In a field filled with overly complicated methods chasing the slightest improvements on metrics, they demonstrate the competitiveness of a simpler approach

**Summary Of Recommendation:**

Experiments section does not compare enough existing methods for me to consider it an "analysis of what matters". Proposed methods are too straightforward to accept as lone contribution.

---

### Author Rebuttal · Authors · 2024-08-08

This includes the updated manuscript and supplemental material based on reviewer feedback. We address each reviewer's feedback with replies to their initial review.

---

### Decision · Program_Chairs · 2024-09-04

**Decision:**

Accept

**Comment:**

Summarizing the strengths and weaknesses pointed out by the reviewers:

Strengths:
- Clear and straightforward premise
- Good ablation analysis and baseline comparisons
- Experimental results across 2 datasets
- Well-written and clear
- The proposed method is simple yet outperforms more complicated baselines

Weaknesses:
- Proposed methods are straightforward and simple, limiting their novelty
- Details of the sub-sampling method are not clearly explained
- Limitations of range-view methods are not well addressed
- Some performance improvements are only effective on the Argoverse dataset

The reviewers disagree on the rigor of the experiments. 1 found them "rigorous and convincing" while the other 2 wanted to see more, particularly against other methods or on other datasets. Since more experiments can't hurt, the authors might consider addressing the concerns of those 2 reviewers by running more experiments.

Overall, 2 reviewers recommended acceptance and 1 rejection, though all weakly. Given the specific feedback of the reviewers, the authors are advised to address their concerns as best as possible.

**Update after rebuttal**: The authors' rebuttal directly addressed many of the reviewers' concerns, including many new experiments. Only 2 of the 3 reviewers responded after the rebuttal. The first upgraded their score from weak reject to weak accept. The second maintained a recommendation of weak accept. The final reviewer, who did not respond, was already recommending weak accept, thus, given all reviewers recommend acceptance, I recommend this paper be accepted.